# Potential Role of the Intratumoral Microbiota in Prognosis of Head and Neck Cancer

**DOI:** 10.3390/ijms242015456

**Published:** 2023-10-22

**Authors:** Masakazu Hamada, Hiroaki Inaba, Kyoko Nishiyama, Sho Yoshida, Yoshiaki Yura, Michiyo Matsumoto-Nakano, Narikazu Uzawa

**Affiliations:** 1Department of Oral & Maxillofacial Oncology and Surgery, Osaka University Graduate School of Dentistry, Suita 565-0871, Japan; kyon.0419@gmail.com (K.N.); narayura630@gmail.com (Y.Y.); uzawa.narikazu.dent@osaka-u.ac.jp (N.U.); 2Department of Pediatric Dentistry, Okayama University Graduate School of Medicine, Dentistry and Pharmaceutical Sciences, Okayama 700-8558, Japan; hinaba@okayama-u.ac.jp (H.I.); syoshida@okayama-u.ac.jp (S.Y.); mnakano@cc.okayama-u.ac.jp (M.M.-N.)

**Keywords:** intratumor microbiome, oral bacteria, *Leptotrichia*, head and neck squamous cell carcinoma, RNA sequencing, TCGA, TCMA

## Abstract

The tumor microbiome, a relatively new research field, affects tumor progression through several mechanisms. The Cancer Microbiome Atlas (TCMA) database was recently published. In the present study, we used TCMA and The Cancer Genome Atlas and examined microbiome profiling in head and neck squamous cell carcinoma (HNSCC), the role of the intratumoral microbiota in the prognosis of HNSCC patients, and differentially expressed genes in tumor cells in relation to specific bacterial infections. We investigated 18 microbes at the genus level that differed between solid normal tissue (*n* = 22) and primary tumors (*n* = 154). The tissue microbiome profiles of *Actinomyces*, *Fusobacterium*, and *Rothia* at the genus level differed between the solid normal tissue and primary tumors of HNSCC patients. When the prognosis of groups with rates over and under the median for each microbe at the genus level was examined, rates for *Leptotrichia* which were over the median correlated with significantly higher overall survival rates. We then extracted 35 differentially expressed genes between the over- and under-the-median-for*-Leptotrichia* groups based on the criteria of >1.5 fold and *p* < 0.05 in the Mann–Whitney U-test. A pathway analysis showed that these *Leptotrichia*-related genes were associated with the pathways of Alzheimer disease, neurodegeneration-multiple diseases, prion disease, MAPK signaling, and PI3K-Akt signaling, while protein–protein interaction analysis revealed that these genes formed a dense network. In conclusion, probiotics and specific antimicrobial therapy targeting *Leptotrichia* may have an impact on the prognosis of HNSCC.

## 1. Introduction

The Cancer Genome Atlas (TCGA) is a large comprehensive cancer genome project initiated in 2006 that aims to catalog and discover major cancer-causing genome alterations through multi-dimensional analyses for the creation of new cancer treatments, diagnostics, and prevention methods for more than 20 cancer types [1,2,3]. In 2015, TCGA profiled cases of head and neck squamous cell carcinoma (HNSCC) to comprehensively characterize genomic alterations [4]. HNSCC includes tumors that arise in the lip, oral cavity, pharynx, larynx, and paranasal sinuses. Treatment depends on the location of the tumors that develop, but the standard treatment for HNSCC is a combination of surgery, radiation therapy, and chemotherapy; the 5-year survival rate is only 40–50% despite advances in treatment [5]. Various studies have since been conducted, including the identification of novel prognostic biomarkers for HNSCC using TCGA [6,7,8,9].

Alcohol and tobacco abuse are the most common etiologies of oral, pharyngeal, and laryngeal cancers. In addition, evidence has accumulated supporting the involvement of the microbiome in the developmental process of HNSCC and its susceptibility to chemoradiotherapy. For example, human papillomavirus (HPV)-positive malignancies have been shown to account for approximately 20% of HNSCCs and 55% of those originating in the oropharynx [10]. Patients with HPV-positive-related oropharyngeal squamous cell carcinoma have a better prognosis than HPV-negative patients when treated with chemoradiotherapy [11,12]. Furthermore, the relationship between HNSCC and oral bacteria, such as the periodontopathogen *Porphyromonas gingivalis*, and the caries pathogen *Streptococcus mutans* was recently investigated. Several oral bacteria have been shown to promote tumor progression [13,14,15]. The relationship between the oral microbiome and HNSCC may also have important implications for the prevention and early detection of HNSCC [16].

Microbes in cancer are rapidly being developed as a potentially powerful new toolkit for improving patient care [17,18]. The intratumoral microbiota affects tumor progression through several mechanisms, including DNA damage, the activation of oncogenic pathways, the induction of immunosuppression, and the metabolism of drugs [18]. The Cancer Microbiome Atlas (TCMA) was recently published and includes the curated tissue-endemic microbial profiles of 3689 unique samples from 1772 patients from five TCGA projects and 21 anatomical sites [16]. This may enable a multi-omics analysis with systematic microbe–host matching and has been actively used in studies on various cancers, such as gastric cancer, colon cancer, and HNSCC [19,20,21].

However, the relationship between the abundance of different bacteria in the intratumor microbiome and the prognosis of HNSCC patients in TCMA database remains unclear. Therefore, the present study investigated the intratumoral microbiota of HNSCC and its prognostic impact on patient survival and examined differentially expressed genes in tumor cells associated with specific bacterial infections.

## 2. Results

### 2.1. Microbiome Profiling of 18 Selected Microbes at the Genus Level

We selected 154 patients who were present in both TCGA and TCMA datasets (Figure 1A), and the following microbes were identified to at least the genus level in TCMA database: *Actinomyces*, *Aggregatibacter*, *Alloprevotella*, *Campylobacter*, *Capnocytophaga*, *Fusobacterium*, *Granulicatella*, *Haemophilus*, *Lactobacillus*, *Leptotrichia*, *Mycoplasma*, *Neisseria*, *Porphyromonas*, *Prevotella*, *Rothia*, *Streptococcus*, *Treponema*, and *Veillonella* (Figure 1B). In solid normal tissue, the population with the highest percentage among the 18 selected microbes was *Prevotella*, followed by *Streptococcus* and *Fusobacterium*. In primary tumors, the population with the highest percentage was *Prevotella*, followed by *Fusobacterium* and *Streptococcus*. We then investigated differences in the 18 microbes at the genus level between the solid normal tissue (*n* = 22) and primary tumors (*n* = 154) of HNSCC patients. The results obtained showed that the tissue microbiome profiles of *Actinomyces*, *Fusobacterium*, and *Rothia* differed between solid normal tissue and primary tumors (Figure 1C).

### 2.2. Prognostic Significance of 18 Selected Microbes at the Genus Level in HNSCC Patients in TCGA Database

We examined the relationship between the 18 selected microbes at the genus level and the prognosis of TCGA-HNSCC patients. Patients were divided into two groups based on the rate of occurrence of a microbe being over and under the median. Differences detected in survival times using the Kaplan–Meier method were analyzed with the generalized Wilcoxon test and log-rank test (Figure 2). The generalized Wilcoxon test and log-rank test showed that among the 18 microbes examined, a rate over the median for *Leptotrichia* (Figure 2J) correlated with significantly higher overall survival rates. A rate over the median for *Campylobacter* (Figure 2D) or *Capnocytophaga* (Figure 2E) correlated with significantly higher overall survival rates only in the generalized Wilcoxon test or log-rank test. In contrast, the generalized Wilcoxon test showed that a rate over the median for *Lactobacillus* (Figure 2I) correlated with lower overall survival rates. On the other hand, rates over the median for *Actinomyces* (Figure 2A), *Aggregatibacter* (Figure 2B), *Alloprevotella* (Figure 2C), *Fusobacterium* (Figure 2F), *Granulicatella* (Figure 2G), *Haemophilus* (Figure 2H), *Mycoplasma* (Figure 2K), *Neisseria* (Figure 2L), *Porphyromonas* (Figure 2M), *Prevotella* (Figure 2N), *Rothia* (Figure 2O), *Streptococcus* (Figure 2P), *Treponema* (Figure 2Q), and *Veillonella* (Figure 2R) were not associated with survival rates.

### 2.3. Relationships between Classical Prognostic Factors and Survival Rates Associated with Leptotrichia in TCGA-HNSCC Patients

Since a rate over the median for *Leptotrichia* (Figure 2J) correlated with significantly higher overall survival rates, the effects of drinking (Figure 3A,B), smoking (Figure 3C,D), HPV status (Figure 3E,F), sex (Figure 3G,H), the presence of lymph node metastasis (Figure 3I,J), and tumor size (Figure 3K,L) on survival rates associated with *Leptotrichia* were examined. Survival curves were not affected by the population of *Leptotrichia* in the absence of drinking (Figure 3B), without lymph node metastasis (Figure 3J), in females (Figure 3H), and in T1–T2 tumors (Figure 3K) but were markedly affected by a rate over the median for *Leptotrichia* with other factors. 

### 2.4. Cox Regression Analysis of Relationships of 18 Selected Microbes at the Genus Level and Classical Prognostic Factors with Survival in TCGA-HNSCC Patients

The 18 selected microbes and their correlations were analyzed in more detail. Univariate and multivariate analyses (Cox proportional hazard model) were performed using the 18 selected microbes and classical risk factors, such as sex, HPV, smoking, age, and TNM stage, as independent variables. In the univariate analysis, *Leptotrichia*_High (vs. Low) (HR = 0.380, 95% CI = 0.215–0.669, *p* = 0.001) correlated with the prognosis of TCGA-HNSCC patients (Table 1). In addition, the multivariate analysis showed that *Leptotrichia*_High (vs. Low) (HR = 0.273, 95% CI = 0.116–0.645, *p* = 0.003) correlated with an improved prognosis in TCGA-HNSCC patients (Table 1).

### 2.5. Extraction of Differentially Expressed Genes between over- and under-the-Median-for-Leptotrichia Groups

We focused on *Leptotrichia* at the genus level, which significantly improved the prognosis of TCGA-HNSCC patients, and investigated whether intratumoral *Leptotrichia* affected the expression of genes in HNSCC cells. The population of *Leptotrichia* at the genus level was divided into two groups: a rate over and under the median. Based on the criteria of >1.5 fold and *p* < 0.05 in the Mann–Whitney U-test, we extracted 35 differentially expressed genes between the over- and under-the-median groups, which included both up- and down-regulated genes in tumor cells (Figure 4A). We produced a heat map to show the up- or down-regulated expression profiles of the 35 differentially expressed genes between the over- and under-the-median groups (Figure 4B).

### 2.6. Functional and Protein–Protein Interaction (PPI) Analyses of Leptotrichia-Related Genes

The 35 differentially expressed genes between the over- and under-the-median-for-*Leptotrichia* groups extracted based on the criteria of >1.5 fold and *p* < 0.05 in the Mann–Whitney U-test were subjected to functional and PPI analyses. Gene Ontology (GO) terms and Kyoto Encyclopedia of Genes and Genomes (KEGG) pathway analyses were performed to investigate biological properties and potential signaling pathways. In the GO enrichment analysis, enriched terms were as follows: the positive regulation of gene expression, the positive regulation of mitotic nuclear division, the positive regulation of cell division, the positive regulation of cytokine production, fever generation, the positive regulation of heart induction via the negative regulation of the canonical Wnt signaling pathway, the positive regulation of cell proliferation, the positive regulation of immature T cell proliferation in the thymus, the positive regulation of interleukin-6 production, the ERBB2-EGFR signaling pathway, the regulation of nitric oxide synthase activity, the positive regulation of prostaglandin secretion, the positive regulation of epidermal growth factor-activated receptor activity, the positive regulation of transcription, DNA-templating, the cytokine-mediated signaling pathway, the positive regulation of keratinocyte proliferation, the positive regulation of angiogenesis, ectopic germ cell programmed cell death, the positive regulation of glial cell proliferation, the negative regulation of cell proliferation, the cellular response to lipopolysaccharide, the negative regulation of apoptotic process, the positive regulation of vascular endothelial growth factor production, and cell–cell signaling (Figure 5A). The KEGG analysis revealed that the 35 extracted genes were significantly enriched in the pathways of Alzheimer disease, the pathways of neurodegeneration-multiple diseases, prion disease, the MAPK signaling pathway, and the PI3K-Akt signaling pathway (Figure 5B). The PPI network analysis showed that AREG, DKK1, EREG, IL1A, IL1B, LAMC2, RAG1, SLC7A5, CDKN2A, CECR2, CYP24A1, HMGN5, MYB, and WNT11 formed a dense network among these genes (Figure 5C). Among these genes, CDKN2A, CECR2, CYP24A1, HMGN5, MYB, and WNT11 were up-regulated with a fold change (over/under the median) > 1.5, while AREG, DKK1, EREG, IL1A, IL1B, LAMC2, RAG1, and SLC7A5 were down-regulated with a fold change < 0.66.

## 3. Discussion

The oral microbiome is a complex ecological environment comprising 750 microorganisms, including bacteria, archaea, protozoa, fungi, and viruses [22,23]. TCMA, which has led to the discovery of prognostic species and blood signatures of mucosal barrier injury and enables multi-omics analyses with systematic microbe–host matching, was recently published [24]. TCMA is also used in HNSCC to search for novel microbial markers and causative bacteria of the inflammatory tumor microenvironment [25,26]. In the present study, 221 microbes were detected at the genus level and were narrowed down to 18 microbes based on their percentages. *Fusobacterium* was found to be among the most abundant species in both normal and tumor tissues, while *Porphyromonas* was among the least abundant species in both normal and tumor tissues (Figure 1). *Fusobacterium* and *Porphyromonas* species, which are associated with the prognosis of cancer, did not affect patient prognosis in an analysis of two classified groups, a rate over and under the median, while the genus *Leptotrichia* was shown to improve patient prognosis (Figure 2). The reason why *Fusobacterium* and *Porphyromonas* species were not clearly associated with the prognosis of HNSCC patients may be related to the amount of bacteria in the selected tissues.

*Leptotrichia* species are biochemically anaerobic Gram-negative rods that belong to the normal flora of humans and are generally present in the oral cavity, intestines, and human female genitalia [27]. *Leptotrichia* were found to be significantly more abundant in allergic rhinitis (AR) and allergic rhinitis with asthma (ARAS) [28]; the composition ratios of *Leptotrichia* species were higher in AR and ARAS (5.9 and 5.2%, respectively) than in healthy controls (3.5%). Another study reported a relationship between the use of dentures and *Candida albicans* [29]. The composition ratios of *Leptotrichia* were 3–4% in dentures and dental plaque, and, thus, the genus *Leptotrichia*, which negatively correlated with *C. albicans*, may be useful in antifungal therapy to control the growth of *C. albicans* [29]. *C. albicans*, the most common oral commensal, is also associated with cancer and has been suggested to exert tumorigenic effects and affect PD-L1 expression [30,31]. In the present study, patients in the over-the-median-for-*Leptotrichia* group had a better prognosis; therefore, *C. albicans* may have been less abundant under that condition.

*Leptotrichia* and *Fusobacterium* have been implicated in the development of colon cancer [25,32]. Furthermore, intratumoral *Leptotrichia* has been identified as a novel microbial marker of a favorable clinical outcome in HNSCC patients [25]. *Leptotrichia* species correlated with higher overall survival rates and were significantly more abundant in early-stage patients than in advanced-stage patients, suggesting the protective effects of *Leptotrichia* species in the HNSCC tumor microenvironment [25]. We herein investigated the effects of classical risk factors, such as drinking, smoking, HPV status, sex, lymph node metastasis, and tumor size, in two groups: a rate over and under the median for *Leptotrichia* species. Survival curves were not affected by the population of *Leptotrichia* species in the absence of drinking, without lymph node metastasis, in females, and in T1–T2 tumors, but were markedly changed when the rate of *Leptotrichia* species was over the median with other classical prognostic factors examined in the present study. The result on tumor sizes is not consistent with previous findings [22]. The present results suggest that the genus *Leptotrichia* was less protective in the HNSCC tumor microenvironment in early-stage patients.

Microbial and molecular differences in HNSCC sites were recently studied [33]. Through KEGG pathway analysis, it was found that, in oral cancers, positively correlated genes were prion diseases, Alzheimer disease, Parkinson disease, *Salmonella* infection, and pathogenic *Escherichia coli* infection. In non-oral cancers, positively correlated genes were herpes simplex virus 1 infection and spliceosome, while negatively correlated genes were the PI3K-Akt signaling pathway, focal adhesion, the regulation of actin cytoskeleton, ECM–receptor interaction, and dilated cardiomyopathy. In the present study, genes differentially expressed between the over- and under-the-median-for-*Leptotrichia* groups were extracted to assess the effects of intratumoral *Leptotrichia* on the behavior of HNSCC (Figure 4). KEGG analysis showed that *Leptotrichia*-related genes were significantly altered in Alzheimer disease, pathways of neurodegeneration - multiple diseases, prion disease, the MAPK signaling pathway, and the PI3K-Akt signaling pathway, consistent with previous reports not limited to the genus *Leptotrichia*. Pathways of neurodegeneration-multiple diseases and MAPK signaling, which were identified for the first time in the present study, may be hallmarks of the effects of *Leptotrichia* on HNSCC. Furthermore, the *Leptotrichia*-related genes selected in the present study, which were significantly up- or down-regulated and formed a dense PPI network, were associated with a favorable prognosis in HNSCC patients. However, bioinformatics-based research on the microbiota and HNSCC is in the early stages, and further research will help to elucidate the prognosis and progression of HNSCC.

## 4. Materials and Methods

### 4.1. Data Collection from TCGA and TCMA Databases

We obtained the RNA-seq count data (HTSeq version) of TCGA-HNSC (499 primary-tumor and 45 solid-tissue normal samples) from the GDC Data Portal [34] (https://portal.gdc.cancer.gov/ accessed on 20 March 2019) with the Subio Platform (https://www.subioplatform.com accessed on 17 October 2023). We also obtained the intratumor microbiome compositions of 177 TCMA-HNSC samples (155 primary tumor and 22 solid tissue normal samples) at the genus level from TCMA [35] (https://tcma.pratt.duke.edu/ accessed on 13 July 2023). We selected 154 patients in both TCGA and TCMA (Figure 1A).

### 4.2. Filtering of TCMA Genus Microbes

We used Subio Platform [36] software v1.24.5859 (Subio Inc. Aichi, Japan) to filter microbes. A total of 221 microbes were defined at TCMA genus level, but most were undetected; therefore, we excluded those with a rate < 0.1 in 175 out of 177 samples. A total of 18 microbes remained, and we summed the rates of the 203 other microbes and labeled them as “Other”.

### 4.3. Kaplan–Meier Survival Analysis

Regarding each of the filter-passed 18 microbes, TCGA-HNSC primary tumor samples were divided into two groups: a rate over and under the median. Subio Platform software was used to generate Kaplan–Meier survival curves for comparisons of the outcomes of the over and under the median groups for each microbe.

### 4.4. Extraction of Differentially Expressed Genes between over and under the Median for Leptotrichia Groups

We normalized RNA-Seq count data at the 90th percentile, replaced all non-zero counts less than 50 with 50, and replaced 0 with 32 as a low signal cut-off. We turned normalized counts to log2 ratios against the average of solid normal tissue samples. We excluded genes if their counts were too low (counts < 50 in all samples) or too stable (log2 ratios between −1 and 1 in all samples.)

We extracted 35 differentially expressed genes between over and under the median for *Leptotrichia* groups based on the criteria of >1.5 fold and *p* < 0.05 in the Mann–Whitney U-test (Figure 4A). We applied hierarchical clustering to 35 genes (Figure 4B).

### 4.5. Functional Pathway and PPI Analyses

The Database for Annotation, Visualization, and Integrated Discovery (DAVID) server was used to examine the molecular pathways of the selected genes for GO terms and KEGG pathways. GO enrichment was performed over three primary levels: cellular components (CC), biological processes (BP), and molecular functions (MF). Based on the STRING online database (https://string-db.org/ accessed on 5 September 2023), these genes were used to establish a PPI network. We then visualized the most significant modules in the PPI network.

### 4.6. Statistical Analyses

Statistical analyses were performed using Student’s *t*-test with Microsoft Excel (Microsoft, Redmond, WA, USA). Results were expressed as the mean ± SD. Differences were considered significant at *p* < 0.05. In the survival analysis shown in Table 1, the hazard ratio (HR) relative to the indicated reference (ref) value, its 95% confidence interval (CI), and the *p*-value (those < 0.05 are indicated in bold) for the Cox hazard model are shown. The HR and its 95% CI were calculated using a Cox regression analysis after the proper evaluation of assumptions of Cox regression models using the survival package.

## 5. Conclusions

We used TCMA and The Cancer Genome Atlas and examined microbiome profiling in HNSCC. The tissue microbiome profiles of *Actinomyces*, *Fusobacterium*, and *Rothia* at the genus level differed between the solid normal tissue and primary tumors of HNSCC patients. However, when the prognosis of groups with rates over and under the median for each microbe at the genus level was examined using a Cox regression analysis and the Kaplan–Meier method, a rate over the median for *Leptotrichia* was found to be correlated with significantly higher overall survival rates in TCGA-HNSCC patients. *Leptotrichia* species-targeted probiotics and specific antibacterial therapy may have an impact on the prognosis of HNSCC.

## Figures and Tables

**Figure 1 ijms-24-15456-f001:**
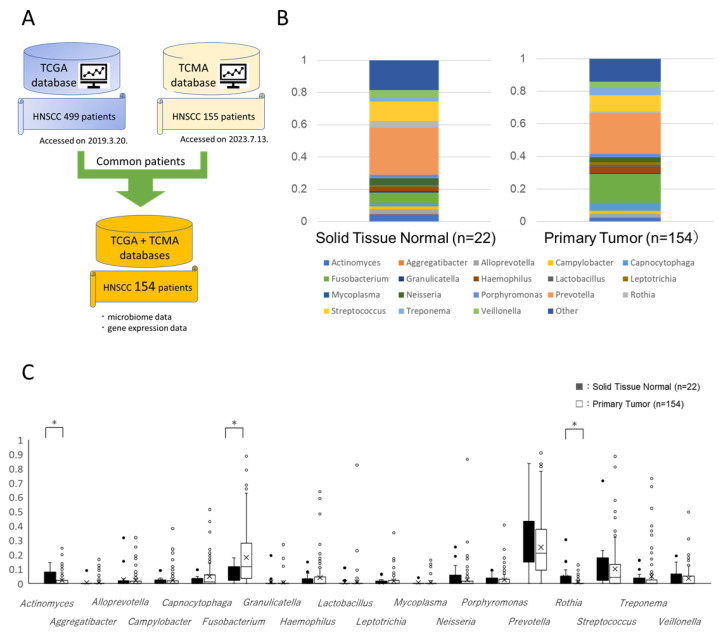
Microbiome profile analysis of 18 selected microbes at the genus level in solid normal tissue and primary tumors of HNSCC patients. (**A**) The extraction schedule of common patients from TCGA and TCMA databases. (**B**) A breakdown of the 18 selected microbes at the genus level, including *Actinomyces*, *Aggregatibacter*, *Alloprevotella*, *Campylobacter*, *Capnocytophaga*, *Fusobacterium*, *Granulicatella*, *Haemophilus*, *Lactobacillus*, *Leptotrichia*, *Mycoplasma*, *Neisseria*, *Porphyromonas*, *Prevotella*, *Rothia*, *Streptococcus*, *Treponema*, and *Veillonella*. (**C**) Comparison of the 18 selected microbes at the genus level in the solid normal tissue and primary tumors of HNSCC patients. * *p* < 0.05.

**Figure 2 ijms-24-15456-f002:**
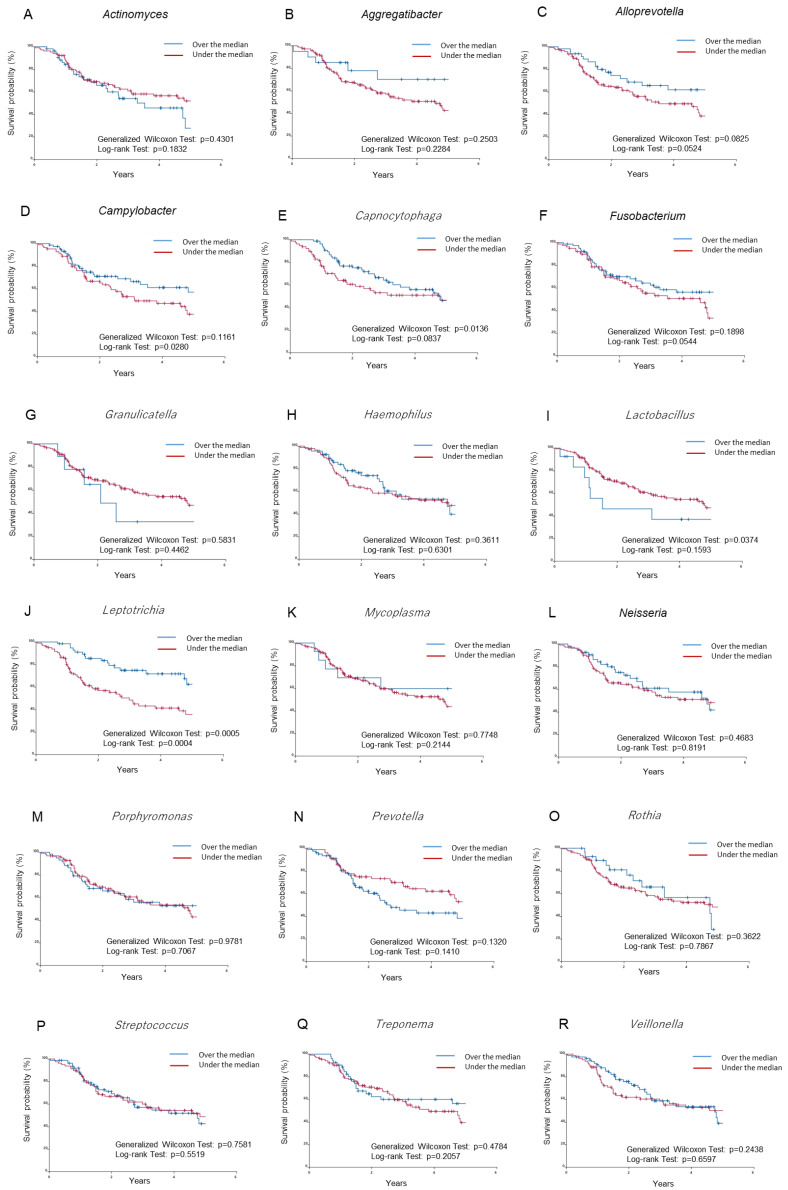
Prognostic significance of 18 selected microbes at the genus level in HNSCC patients in TCGA database. The relationships between the overall survival of TCGA-HNSCC patients and the 18 selected microbes at the genus level were assessed using the Kaplan–Meier method. Differences detected in survival times were then analyzed with the generalized Wilcoxon test and log-rank test. (**A**): *Actinomyces*, (**B**): *Aggregatibacter*, (**C**): *Alloprevotella*, (**D**): *Campylobacter*, (**E**): *Capnocytophaga*, (**F**): *Fusobacterium*, (**G**): *Granulicatella*, (**H**): *Haemophilus*, (**I**): *Lactobacillus*, (**J**): *Leptotrichia*, (**K**): *Mycoplasma*, (**L**): *Neisseria*, (**M**): *Porphyromonas*, (**N**): *Prevotella*, (**O**): *Rothia*, (**P**): *Streptococcus*, (**Q**): *Treponema*, (**R**): *Veillonella*. Differences were considered significant at *p* < 0.05.

**Figure 3 ijms-24-15456-f003:**
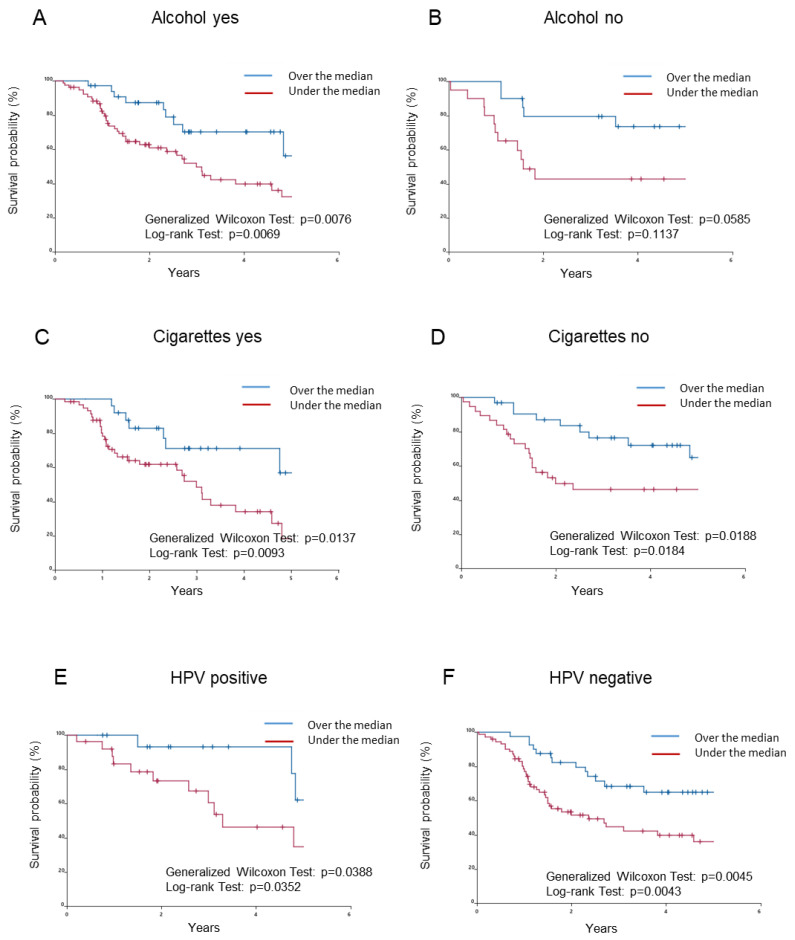
Relationships between classical prognostic factors and survival rates associated with *Leptotrichia* in TCGA-HNSCC patients. Survival curves were recalculated based on the population of *Leptotrichia* in consideration of classical prognostic factors, such as drinking, smoking, HPV status, sex, the presence of lymph node metastasis, and tumor sizes. (**A**) A history of drinking. (**B**) No history of drinking. (**C**) A history of smoking. (**D**) No history of smoking. (**E**) A history of HPV infection. (**F**) No history of HPV infection. (**G**) Males. (**H**) Females. (**I**) Lymph node metastasis. (**J**) No lymph node metastasis. (**K**) Tumor sizes T1–T2. (**L**) Tumor sizes ≥ T3. Differences were considered significant at *p* < 0.05.

**Figure 4 ijms-24-15456-f004:**
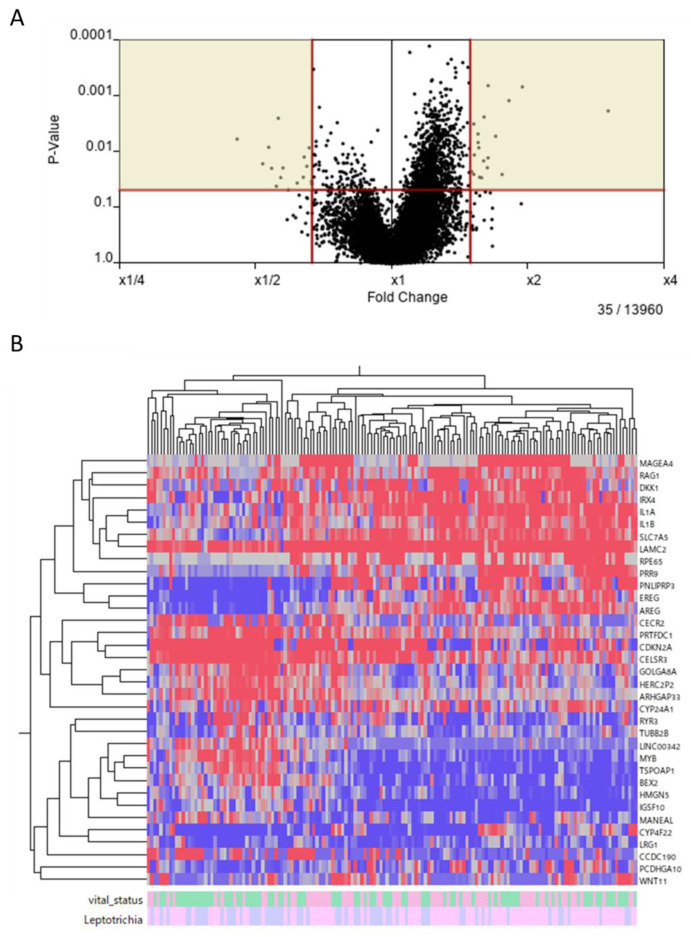
Extraction of genes related to *Leptotrichia.* (**A**) Based on the criteria of >1.5 fold and *p* < 0.05 in the Mann–Whitney U-test, 35 differentially expressed genes between the over- and under-the-median-for-*Leptotrichia* groups were extracted. (**B**) Heat map and hierarchical clustering of the 35 extracted genes. Colors from blue to red indicate low to high expression levels. The vital status of TCGA-HNSCC patients and the rate of *Leptotrichia* are color-coded as follows: alive (■), dead (■), over the median (■), and under the median (■), respectively.

**Figure 5 ijms-24-15456-f005:**
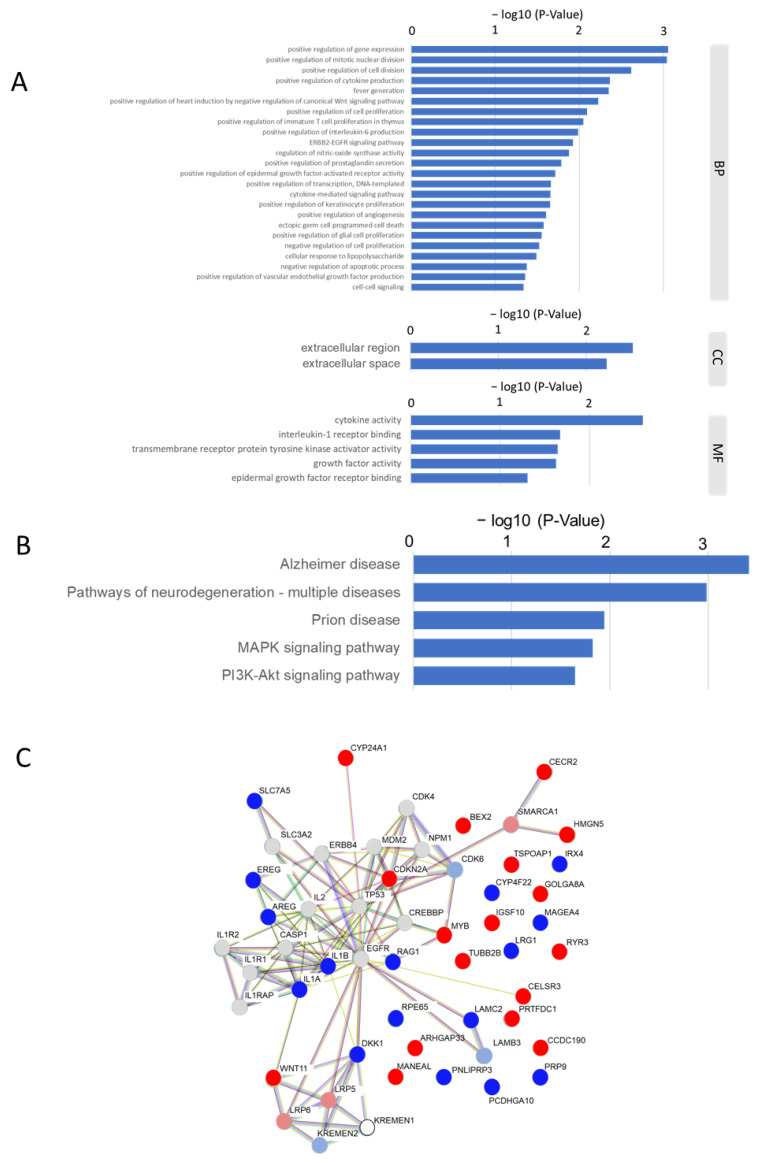
Functional and protein–protein interaction analyses of *Leptotrichia*-related genes. (**A**) A list of GO terms identified through the GO enrichment analysis of 35 differentially expressed genes between the over and under the median for *Leptotrichia* groups extracted based on the criteria of >1.5 fold and *p* < 0.05 in the Mann–Whitney U-test. BP, biological process; CC, cellular composition; MF, molecular function. (**B**) List of molecular pathways identified via the KEGG pathway enrichment analysis of the 35 extracted genes. (**C**) Proteins encoded by the 35 extracted genes were subjected to a PPI network analysis. Genes with fold changes (over/under the median) > 1.5, 1.5 > fold change > 1.2, 1.2 > fold change > 0.83, 0.83 > fold change > 0.66, and 0.66 > fold change are color-coded as red (■), pale red (■), gray (■), pale blue (■), and blue (■), respectively. White color means none of the above.

**Table 1 ijms-24-15456-t001:** Univariate and multivariate analyses of the relationship of the 18 microbes at the genus level and classical prognostic factors with survival in TCGA-HNSCC patients.

		Univariate					Multivariate				
		HR	95% CI			*p*-Value	HR	95% CI			*p*-Value
*Actinomyces*_High (vs. Low)		1.350	0.813	-	2.240	0.246	4.090	1.553	-	10.774	**0.004**
*Aggregatibacter*_High (vs. Low)		0.508	0.204	-	1.266	0.146	0.546	0.202	-	1.478	0.234
*Alloprevotella*_High (vs. Low)		0.578	0.324	-	1.030	0.063	0.620	0.283	-	1.359	0.233
*Campylobacter*_High (vs. Low)		0.644	0.389	-	1.067	0.088	0.560	0.291	-	1.079	0.083
*Capnocytophaga*_High (vs. Low)		0.714	0.438	-	1.162	0.175	0.766	0.392	-	1.496	0.435
*Fusobacterium*_High (vs. Low)		0.722	0.442	-	1.178	0.192	1.446	0.701	-	2.981	0.318
*Granulicatella*_High (vs. Low)		1.503	0.602	-	3.750	0.383	5.873	1.495	-	23.079	**0.011**
*Haemophilus*_High (vs. Low)		0.890	0.536	-	1.476	0.651	0.809	0.373	-	1.753	0.590
*Lactobacillus*_High (vs. Low)		1.747	0.796	-	3.832	0.164	1.463	0.407	-	5.259	0.560
*Leptotrichia*_High (vs. Low)		0.380	0.215	-	0.669	**0.001**	0.273	0.116	-	0.645	**0.003**
*Mycoplasma*_High (vs. Low)		0.758	0.303	-	1.895	0.554	1.305	0.437	-	3.891	0.633
*Neisseria*_High (vs. Low)		0.880	0.523	-	1.480	0.630	1.429	0.677	-	3.016	0.349
*Porphyromonas*_High (vs. Low)		0.978	0.591	-	1.618	0.931	1.435	0.765	-	2.692	0.260
*Prevotella*_High (vs. Low)		1.589	0.971	-	2.600	0.065	2.378	1.123	-	5.034	**0.024**
*Rothia*_High (vs. Low)		0.876	0.457	-	1.677	0.689	1.114	0.375	-	3.313	0.846
*Streptococcus*_High (vs. Low)		1.037	0.637	-	1.689	0.883	0.723	0.322	-	1.621	0.430
*Treponema*_High (vs. Low)		0.805	0.475	-	1.366	0.422	0.792	0.337	-	1.863	0.593
*Veillonella*_High (vs. Low)		0.915	0.562	-	1.492	0.722	0.453	0.183	-	1.121	0.087
Age (per 1 year)		1.000	0.979	-	1.021	0.993	1.022	0.991	-	1.055	0.168
Sex_male (vs. female)		0.825	0.491	-	1.388	0.469	0.936	0.464	-	1.889	0.854
HPV status_Positive (vs. Negative)		0.693	0.384	-	1.253	0.225	0.499	0.206	-	1.208	0.123
Alcohol_history_Yes (vs. No)		1.302	0.736	-	2.304	0.365	1.054	0.460	-	2.410	0.902
Cigarettes per day_>0 (vs. 0)		1.308	0.797	-	2.146	0.288	1.069	0.547	-	2.091	0.845
M stage_m1 (vs. m0)		8.793	1.166	-	66.316	**0.035**	45.367	1.018	-	2022.711	**0.049**
N stage (Continuous variable per 1)		1.093	0.966	-	1.236	0.158					
N stage (Category)											
	n0	1.000	ref				1.000		ref		
	n1	1.121	0.563	-	2.230	0.746	1.855	0.769	-	4.472	0.169
	n2	2.924	1.119	-	7.640	**0.029**	3.238	0.684	-	15.332	0.139
	n2a	2.774	0.835	-	9.215	0.096	3.365	0.780	-	14.517	0.104
	n2b	0.767	0.332	-	1.768	0.533	1.328	0.468	-	3.767	0.594
	n2c	2.360	1.097	-	5.077	**0.028**	2.215	0.822	-	5.967	0.116
	n3	2.110	0.499	-	8.926	0.310	4.001	0.400	-	39.979	0.238
T stage (Continuous variable per 1)		1.160	0.895	-	1.503	0.262					
T stage (Category)											
	t1	1.000		ref			1.000		ref		
	t2	0.852	0.252	-	2.882	0.797	1.542	0.264	-	9.025	0.631
	t3	0.912	0.264	-	3.155	0.884	1.988	0.319	-	12.393	0.462
	t4a	1.198	0.363	-	3.952	0.767	2.101	0.345	-	12.794	0.421
	t4b	1.752	0.181	-	16.908	0.628	1.760	0.034	-	91.851	0.779

HR: hazard ratio; 95% CI: 95% confidence interval; ref: reference value. The hazard ratio refers to a high/low survival. A multivariate analysis was performed with forced insertion of all variables. Bold type indicates *p* < 0.05.

## Data Availability

Data are available from the corresponding author upon reasonable request.

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
