# Peer review of "Potential Role of the Intratumoral Microbiota in Prognosis of Head and Neck Cancer"

_ijms, 2023, doi:10.3390/ijms242015456_

Round 1
Reviewer 1 Report
The text reads well in general. It is not clear (at least to me) how treatment influence was statistically tackled with. Please clarify it. In this context, in the introduction, it should be stressed HNSCC treatment strategy. Therefore, introduction would be enhanced by addition of references, such as PMID: 31097964 and ID: 27185284, to better contextualize the issue at hand in oncologic scenario. Please add details on “patients at risk” in all survival curves.
Author Response
The text reads well in general. It is not clear (at least to me) how treatment influence was statistically tackled with. Please clarify it. In this context, in the introduction, it should be stressed HNSCC treatment strategy. Therefore, introduction would be enhanced by addition of references, such as PMID: 31097964 and ID: 27185284, to better contextualize the issue at hand in oncologic scenario. Please add details on “patients at risk” in all survival curves.
Response: Thank you for your positive comments to our paper. We followed the reviewer’s suggestion and added the following sentences “HNSCC includes tumors that arise in the lip, oral cavity, pharynx, larynx, and para-nasal sinuses. Treatment depends on the location of the tumors that develop, but the standard treatment for HNSCC is a combination of surgery, radiation therapy, and chemotherapy; the 5-year survival rate is only 40-50% despite advances in treatment [5]” (lines 39-43, ref 5). “Alcohol and tobacco abuse are the most common etiologies of oral, pharyngeal, and laryngeal cancers. In addition, evidence has accumulated supporting the involvement of the microbiome in the developmental process of HNSCC and its susceptibility to chemoradiotherapy. For example, human papillomavirus (HPV)-positive malignancies have been shown to account for approximately 20% of HNSCC and 55% of those originating in the oropharynx [10]. Patients with HPV positive related oropharyngeal squamous cell carcinoma have a better prognosis than HPV-negative patients when treated with chemoradiotherapy [11,12]. Furthermore, the relationship between HNSCC and oral bacteria, such as the periodontopathogen Porphyromonas gingivalis, and the caries pathogen Streptococcus mutans was recently investigated. Several oral bacteria have been shown to promote tumor progression [13-15]. The relationship between the oral microbiome and HNSCC may also have important implications for prevention and early detection of HNSCC [16]” (lines 45-57, ref 10-12).
Reviewer 2 Report
This study aimed at investigating the role of the intratumoral microbiota in the prognosis of patients with head and neck squamous cell carcinoma and the pattern of differentially expressed genes in tumor cells related to bacterial species using the Cancer Microbiome Atlas and Cancer Genome Atlas databases. Overall, the manuscript is clear and well written. However, some points need to be clarified before considering the paper for publication.
Figures 2,3. The level of statistical significance should be added to the footnote. If p=0.05, it is unclear why “Survival curves were not affected with lymph node metastasis (Generalizes Wilcoxon test: p=0.006; Log-rang test: p=0.006). You probably mean “without lymph node metastasis”?
Lines 136-137. Why did the authors say this? When was this data shown?
Table 1. The title appears incomplete. Does the hazard ratio refer to a high/low survival? Furthermore, the confounders considered in the multivariate model should be listed in the footnote.
Lines 216-218. Could the authors provide more details to this statement? Fusobacterium, for instance, was found to be among the most abundant species both in normal and tumor tissues. What about Porphyromonas species?
Lines 242-243. “…but were markedly changed when the rate of Leptotrichia species was over the median 242 with other factors”. Please clarify this point.
Minor editing of English language required
Author Response
This study aimed at investigating the role of the intratumoral microbiota in the prognosis of patients with head and neck squamous cell carcinoma and the pattern of differentially expressed genes in tumor cells related to bacterial species using the Cancer Microbiome Atlas and Cancer Genome Atlas databases. Overall, the manuscript is clear and well written. However, some points need to be clarified before considering the paper for publication.
Response: Thank you for your positive comments to our paper.
Figures 2,3. The level of statistical significance should be added to the footnote. If p=0.05, it is unclear why “Survival curves were not affected with lymph node metastasis (Generalizes Wilcoxon test: p=0.006; Log-rang test: p=0.006). You probably mean “without lymph node metastasis”? 
Response: Following the suggestion of the reviewer, we added a sentence “Differences were considered significant at P<0.05”. to the footnote in Figure 2,3 (lines 121, 141-142). This time, we found that line 130 was incorrect. We changed “with lymph node metastasis” to “without lymph node metastasis” (line 130).
Lines 136-137. Why did the authors say this? When was this data shown?
Response: The sentence has been changed to “The 18 selected microbes correlated were analyzed in more detail” (line 146).
Table 1. The title appears incomplete. Does the hazard ratio refer to a high/low survival? Furthermore, the confounders considered in the multivariate model should be listed in the footnote.
Response: We have changed the subtitle to “Univariate and multivariate analyses of the relationship of the 18 microbes at the genus level and classical prognostic factors with survival in TCGA-HNSCC patients” (lines 155-156). We also added the sentences “the hazard ratio refers to a high/low survival” (line 157) and “A multivariate analysis was performed with forced insertion of all variables” (line 158) to the footnote in Table 1. 
Lines 216-218. Could the authors provide more details to this statement? Fusobacterium, for instance, was found to be among the most abundant species both in normal and tumor tissues. What about Porphyromonas species?
Response: We added ‘Fusobacterium was found to be among the most abundant species in both normal and tumor tissues, while Porphyromonas was among the least abundant species in both normal and tumor tissues (Fig. 1.).’ This has been added to the text (lines 224-226).
Lines 242-243. “…but were markedly changed when the rate of Leptotrichia species was over the median 242 with other factors”. Please clarify this point.
Response:Following the suggestion of the reviewer, we have added the sentences “Survival curves were not affected by the population of Leptotrichia species in the absence of drinking, without lymph node metastasis, in females, and in T1-T2 tumors, but were markedly changed when the rate of Leptotrichia species was over the median with other classical prognostic factors examined in the present study” (lines 253-256).
Reviewer 3 Report
In this work, the authors examined microbiome profiling in head and neck squamous cell carcinoma through Cancer Microbiome Atlas (TCMA) and Cancer Genome Atlas database and disclosed that probiotics and specific antimicrobial therapy targeting Leptotrichia may have an impact on the prognosis of head and neck squamous cell carcinoma (HNSCC). This topic is interesting and important to provide the basis for the development of new approaches to the treatment of HNSCC. This work have been conducted detailed analysis but some minor problems need to be addressed before acceptance. It is suggested to leave two blank spaces at the beginning of line 116, 136,150,168, etc.
In this work, the authors examined microbiome profiling in head and neck squamous cell carcinoma through Cancer Microbiome Atlas (TCMA) and Cancer Genome Atlas database and disclosed that probiotics and specific antimicrobial therapy targeting Leptotrichia may have an impact on the prognosis of head and neck squamous cell carcinoma (HNSCC). This topic is interesting and important to provide the basis for the development of new approaches to the treatment of HNSCC. This work have been conducted detailed analysis but some minor problems need to be addressed before acceptance. It is suggested to leave two blank spaces at the beginning of line 116, 136,150,168, etc.
Author Response
In this work, the authors examined microbiome profiling in head and neck squamous cell carcinoma through Cancer Microbiome Atlas (TCMA) and Cancer Genome Atlas database and disclosed that probiotics and specific antimicrobial therapy targeting Leptotrichia may have an impact on the prognosis of head and neck squamous cell carcinoma (HNSCC). This topic is interesting and important to provide the basis for the development of new approaches to the treatment of HNSCC. This work have been conducted detailed analysis but some minor problems need to be addressed before acceptance. It is suggested to leave two blank spaces at the beginning of line 116, 136,150,168, etc.
Response:Thank you for your positive comments to our paper. We left two spaces at the beginning of lines 125, 146, 161, 281, 290, 296, 303, 313, 321.
Round 2
Reviewer 1 Report
revision ok